# Tracing the Repertoire of Promiscuous Enzymes along the Metabolic Pathways in Archaeal Organisms

**DOI:** 10.3390/life7030030

**Published:** 2017-07-13

**Authors:** Mario Alberto Martínez-Núñez, Zuemy Rodríguez-Escamilla, Katya Rodríguez-Vázquez, Ernesto Pérez-Rueda

**Affiliations:** 1Laboratorio de Estudios Ecogenómicos, Facultad de Ciencias, Unidad Académica de Ciencias y Tecnología de la UNAM en Yucatán, Universidad Nacional Autónoma de México, Carretera Sierra Papacal-Chuburna Km. 5, C.P. 97302, Mérida, Yucatán, Mexico; 2Departamento de Microbiología, Instituto de Biotecnología, Universidad Nacional, Autónoma de México, C.P. 62210, Cuernavaca, Morelos, Mexico; zuemy.rodriguez@gmail.com; 3Instituto de Investigaciones en Matemáticas Aplicadas y en Sistemas, Universidad Nacional Autónoma de México, Ciudad Universitaria, C.P. 04510, Ciudad de México, Mexico; katya.rodriguez@iimas.unam.mx; 4Departamento de Ingeniería Celular y Biocatálisis, Instituto de Biotecnología, Universidad Nacional Autónoma de México, C.P. 62210, Cuernavaca, Morelos, Mexico; erueda@ibt.unam.mx; 5Instituto de Investigaciones en Matemáticas Aplicadas y en Sistemas, Universidad Nacional Autónoma de México, Unidad Académica Yucatán, Carretera Sierra Papacal-Chuburna Km. 5, C.P. 97302, Mérida, Yucatán, Mexico

**Keywords:** promiscuous enzymes, Enzyme Commission number, metabolism, archaea, comparative genomics

## Abstract

The metabolic pathways that carry out the biochemical transformations sustaining life depend on the efficiency of their associated enzymes. In recent years, it has become clear that promiscuous enzymes have played an important role in the function and evolution of metabolism. In this work we analyze the repertoire of promiscuous enzymes in 89 non-redundant genomes of the Archaea cellular domain. Promiscuous enzymes are defined as those proteins with two or more different Enzyme Commission (E.C.) numbers, according the Kyoto Encyclopedia of Genes and Genomes (KEGG) database. From this analysis, it was found that the fraction of promiscuous enzymes is lower in Archaea than in Bacteria. A greater diversity of superfamily domains is associated with promiscuous enzymes compared to specialized enzymes, both in Archaea and Bacteria, and there is an enrichment of substrate promiscuity rather than catalytic promiscuity in the archaeal enzymes. Finally, the presence of promiscuous enzymes in the metabolic pathways was found to be heterogeneously distributed at the domain level and in the phyla that make up the Archaea. These analyses increase our understanding of promiscuous enzymes and provide additional clues to the evolution of metabolism in Archaea.

## 1. Introduction

The adaptation of organisms to environment changes depends on the cellular adequacy to abiotic and biotic variables, such as temperature, nutrients availability, salinity, or the interaction with others organisms, which are in constant changes. In this context, diverse cellular components such as transporters, regulatory proteins and enzymes converge to contend and adapt against these changes. The role that enzymes play in these processes is fundamental, since they achieve the biochemical transformations of substrates into useful products, providing the cell with matter and energy necessary to grow. Enzymes have usually been described by catalyzing only one reaction on specific substrates, thus they are commonly considered as “specialists”; however, some enzymes may also be multifunctional [1,2]. Multifunctional enzymes have been defined as proteins playing multiple physiological roles in the cell, and can be classified as moonlighting and promiscuous. Moonlighting enzymes are usually composed by a structural domain that performs the catalytic activity, and a non-catalytic domain, associated with regulation or protein–protein interactions, among others activities; in counterpart, promiscuous enzymes are only composed of catalytic domains performing several biochemical functions [3,4,5,6]. Promiscuous enzymes can be further classified into substrate promiscuous enzymes, i.e., enzymes with relaxed or broad substrate specificity; and catalytic promiscuous enzymes, i.e., enzymes catalyzing distinctly different chemical transformations with different transition states [7,8]. In this context, diverse mechanisms that allow enzymatic promiscuity, such as conformational diversity, alternative substrates positions, and different protonation states, among others have been described [9]. Together, these functional and evolutionary processes associated with enzymes shape the evolution of metabolic systems [10,11].

From a genomic perspective, it has been reported that around 10% of the total enzymatic repertoire in bacterial and archaeal organisms corresponds to promiscuous enzymes [12]. In this regard, taking into account the lower number of genes on archaeal genomes, whose genome sizes range from ~0.5 Mb in the case of *Nanoarchaeum equitans* [13] to ~5.5 Mb in *Methanosarcina barkeri* [14], it has been proposed that archaeal enzymes show a greater degree of multifunctionality [2]. In order to have a more detailed perspective of the multifunctionality of archaeal enzymes, we evaluated in a systematic and genomic way the content and the structural domains in the enzymatic repertoire of archaeal genomes, and in particular, those involving the promiscuous enzymes, as well as the distribution of promiscuous enzymes on metabolic pathways. To achieve this analysis, the content of promiscuous enzymes and their structural domains was accomplished in 89 non-redundant archaeal genomes. Based on statistical evaluations, it was identified that the distribution of promiscuous enzymes is lower in Archaea than in Bacteria. Another outstanding result of our analysis is the greater structural diversity that promiscuous enzymes exhibit in comparison to specialized enzymes, both in Archaea and Bacteria. We also found that the presence of promiscuous enzymes in the metabolic pathways was heterogeneous, both at the domain level, as in phyla associated with Archaea. Finally, we found that the substrate promiscuity is greater than catalytic promiscuity in the archaeal enzymatic repertoire.

## 2. Material and Methods

### 2.1. Genomes and Proteomes Analyzed

89 archaeal and 705 bacterial non-redundant genomes analyzed in this work were retrieved from the National Center for Biotechnology Information (NCBI) ftp server public database section genomes (ftp://ncbi.nlm.nih.gov). These genomes were defined as non-redundant, to exclude any bias associated with overrepresentation of species or strains, as has previously been reported [12]. In brief, through the concatenation of 21 conserved proteins across diverse sequenced genomes, a single data set for phylogenetic analysis was constructed [15]. By eliminating genomes located more closely together on a phylogenetic tree, 794 genomes phylogenetically distant from each other were obtained (Appendix A). We considered only genes with open reading frames (ORFs) that encode predicted protein sequences. 

### 2.2. Identification of Enzymes

For each protein sequence, the annotation of an Enzyme Commission number (E.C. number) using the Kyoto Encyclopedia of Genes and Genomes (KEGG) database was associated [16]. Then, for each enzyme associated with an E.C. number, the assignments of structural domains at sequence level were determined with the Superfamily database [17]. Our inclusion criterion was quite strict because we were only interested in enzymatic sequences with identified metabolic contexts and structural assignments. Finally, promiscuous enzymes were identified in this set of filtered data as those protein sequences containing two or more different E.C. numbers (Appendix A). All processes and data management were conducted using ad hoc Perl scripts.

#### Accuracy of the Promiscuous Enzymes

In order to evaluate our operational definition of promiscuous enzymes, the enzyme promiscuity prediction server was used [18]. In this regard, a representative dataset of 400 out of 51,572 promiscuous enzymes was selected by using the formula (N × Z^2^ × *p* × *q*)/((NE^2^) + (Z^2^ × *p* × *q*)), with a margin of error of 4.9% and a confidence level of 95% (N = population size, Z = confidence level, E = sample error desired; *p* and *q* are the probability to be promiscuous or not, and were assumed to be 0.5). In addition, 400 out 19,411 promiscuous enzymes (positive dataset), and 400 of 48,846 non-promiscuous enzymes (negative control) provided by Carbonell and Faulon were also evaluated. Based on these comparisons, we calculated the: (1) true positives (TP): promiscuous enzymes with more than two different E.C. numbers, and structural domains (SUPFAM) (our dataset) and identified by the Promis server, and Carbonell and Faulon’s positive set identified by the Promis server; (2) false positives (FP): proteins identified as promiscuous enzymes with the Promis server from the Carbonell and Faulon’s negative dataset; (3) false negatives (FN): promiscuous enzymes (our dataset) and Carbonell and Faulon’s positive dataset, identified as negative with the Promis server; (4) sensitivity, Sn = TP/(TP + FN), is the fraction of promiscuous enzymes identified with the Promis server; (5) positive predictive value, PPV = TP/(TP + FP), is the fraction of the promiscuous enzymes inferred; (6) Accuracy, Ac = (Sn + PPV)/2, is the PPV and Sn average.

### 2.3. Statistical Analysis

Non-parametric Wilcoxon rank sum statistical tests were used to evaluate promiscuous enzymes distributions in each archaeal and bacterial domain. Statistical significance was set at *p* ≤ 0.05. The implementation of these tests was carried out using the package *stats* of R programming language for statistical analysis [19]. Data managements were conducted with ad hoc Perl scripts. 

## 3. Results and Discussion

### 3.1. Archaea Exhibit a Lower Promiscuous Enzyme Fraction than Bacterial Genomes

In order to compare the total repertoire of known enzymatic proteins in the genomes of the Archaea, the content of these proteins was exhaustively evaluated. From this analysis, we observed that the 89 genomes, with an average size of 2291 ORFs, contain an average of 487 enzymes. To compare the enzymatic content found in the archaeal genomes, we analyzed 705 bacterial genomes; where, an average of 757 enzymes and an average genome size of 3265 ORFs was observed, i.e., the average content of known enzymes in archaeal genomes is 1.5 times lower than that observed in bacterial genomes. When these values were normalized to obtain the percentage of known enzymes with respect to all the genes of each genome, an average of 22% was found for Archaea, while an average value of 25% was found for Bacteria (Figure 1). The difference between the percentages of enzyme content was found as statistically significant (Wilcoxon test, *p*-value = 4.52 × 10^−6^), showing a higher proportion of known enzymes in Bacteria, with around one-quarter of their genes encoded for enzymes than in Archaea, the latter with one-fifth of the genes encoded for enzymes.

To evaluate whether there is an increase in the number of promiscuous enzymes in the repertoire of known total enzymes previously described, and if this increase might compensate the low proportion of enzymes in Archaea in comparison to Bacteria, the content of promiscuous enzymes was also analyzed. In operational terms, promiscuous enzymes were defined as those enzymes that having two or more different EC numbers, according to KEGG database. In this regard, we found an accuracy of 63% in our identification of promiscuous enzymes in our data according to the evaluation done with the Promis Server [18], suggesting that our definition of promiscuous enzymes is, in general, informative enough. We obtained the ratio between the number of promiscuous enzymes among the total enzymes for each genome. In total, an average value of 8.31% of promiscuous enzymes was found in archaeal genomes against the 8.76% found in bacterial genomes. Although the average contents of promiscuous enzymes show similar values in Archaea and Bacteria, there is a statistically significant difference (Wilcoxon test, *p*-value = 0.03), with a greater proportion of promiscuous enzymes in the bacterial genomes (Figure 1), suggesting that promiscuous enzymes do not compensate the apparent deficit of enzymes observed in Archaea. Therefore, the abundance of enzymes that carry out only one reaction or recognize only one substrate, i.e., specialists, in Archaea could maintain high metabolic fluxes, as has been suggested in *Escherichia coli* [1], necessary to contend against extreme environments, in which the available carbon sources need to be maximized. However, the role of promiscuous enzymes must be also considered as an important source of variability associated with Archaeal metabolism.

### 3.2. Structural Diversity in Promiscuous Enzymes is Higher than Specialist Enzymes

To gain insights into the diversity of promiscuous enzymes identified in Archaea, the structural domains associated with the enzymatic repertoire was evaluated using the protein domains assignments of Superfamily database [17]. From this evaluation, we identified an average number of 60 structural domains in archaeal promiscuous enzymes per genome, decreasing to 48 when non-redundant domains were considered. To obtain the fraction of different superfamily domains, we calculate the ratio between unique structural domains against the total structural domains per genome, an average proportion of 82% of unique structural domains in promiscuous enzymes was found. In order to compare the number and percentage of domains present in the promiscuous enzymes, the same estimates were performed for the structural domains present in the specialist enzymes. The average number of all structural domains was 641, while the average number of different structural domains was 266 in specialist enzymes. The average proportion of unique structural domains in specialist enzymes was 42%. This last value presents a statistically significant difference (Wilcoxon test, *p*-value = 2.2 × 10^−16^) with the fraction of unique structural domains in promiscuous enzymes, observing a greater proportion of different domains in the promiscuous enzymes than in specialists, 82% and 42%, respectively. This significant difference is observed not only in the promiscuous and specialists enzymes of archaeal genomes, but also in the enzymes of bacterial genomes where the fraction of unique domains is greater in promiscuous (73%) than in specialist enzymes (36%) (Wilcoxon test, *p*-value = 2.2 × 10^−16^). In both cases, Archaea and Bacteria, the fraction of non-redundant domains in promiscuous enzymes is twice that in specialist enzymes, showing a greater structural diversity in the enzymes that perform more than one biochemical reaction. Therefore, the biochemical plasticity of the promiscuous enzymes, both in the reactions and in the recognition of substrates, could be as a consequence of their greater structural diversity. In this regard, the expansion of metabolic networks in the evolution of prokaryotes has been dominated by broadening the reaction repertoire [20] or substrate repertoire, rather than by an increase in the number of enzymes encoded in the prokaryotic genomes [12].

### 3.3. Functional Structural Diversity in Promiscuous Enzymes Exhibits a Non-Homogeneous Distribution

To determine the structural diversity associated with promiscuous enzymes, the set of 260 non-redundant structural domains identified in the promiscuous enzymes of all archaeal genomes was evaluated. To this end, archaea was analyzed in terms of their main phyla, finding that Euryarchaeota contain 93% of these domains, followed by Crenarchaeota with 57.6%, and finally Nanoarchaeota with 4.2% (Figure 2). Eleven domains were identified as common to the three phyla, while 121 domains were common to Euryarchaeota and Crenarchaeota. It is interesting to note that all the structural domains identified in Nanoarchaeota were identified as common to all archaea (Figure 2); however, this should be taken with caution because of the low number of Nanoarchaeota genomes represented in our study.

In this regard, the NAD(P)-binding Rossmann-fold (Superfamily ID: 51735) and P-loop containing nucleoside triphosphate hydrolases (Superfamily ID: 52540) domains considered as ancient folds [21], were identified in all three phyla as the most abundant structural domains. The NAD(P)-binding Rossmann-fold domain has also been identified as the largest structural family, and associated with almost all different proteins in the metabolic pathways of small molecules of *Escherichia coli* [22]. In accordance with the diversity of functions associated with P-loop and NAD(P)-binding Rossmann-fold, they have been also associated with the functional metabolic category referred to as “General” functions, according to the classification of the Superfamily database, i.e., associated with multiple functions or interactions with proteins, ions, lipids, or small molecules. The most represented functional categories associated with the most abundant domains corresponds to “Metabolism”, such as Ribokinase and PLP-dependent transferases; “Information”, with domains such as Glutamyl-tRNA (Gln) amidotransferase and DNA topoisomerase; and “Others” with the domain GatB-YqeY motif (Figure 2). In contrast, the less abundant domains corresponds to the Aldehyde ferredoxin oxidoreductase and alpha/beta-Hydrolases, associated with the “Metabolism” functional category. Only one domain with functional category of “Regulation” was found in the GAF domain. This domain has been associated with ligand-binding, and signal transduction [23] and has been found in archaeal regulators, such as the Bat transcriptional regulator of the *Halobacterium* sp. NRC-1 responsible for coordinate regulation of bacteriorhodopsin production [24]. The DPP6 N-terminal (Superfamily ID: 82171), and Periplasmic binding protein-like II (Superfamily ID: 53850) domains identified as less abundant were associated with “Intracellular processes”, such as cell motility/division, and intracellular transport, among others. Finally, the exclusive domains associated with Euryarchaeota represents the 42% of all counted domains, such as the Phosphoglycerate mutase-like or Purine and uridine phosphorylases domains, which belong to the functional metabolic category of “Metabolism”; meanwhile, Crenarchaeota correspond to 6.9%, with domains like the Trypsin-like serine proteases, and S-adenosylmethionine decarboxylase**.** Finally, 12 structural domains were only identified in promiscuous enzymes (Table 1), with a non-homogeneous distribution between the three phyla. This structural diversity identified in promiscuous enzymes might provide functional diversity in archaeal metabolism, increasing the participation of these enzymes in a wide number of metabolic pathways, such as the phosphomannomutase alpha-phosphoglucomutase (EC:5.4.2.8, 5.4.2.2) of *Alicyclobacillus acidocaldarius*, that acts in the pentose phosphate pathway and fructose and mannose metabolisms; the 3-hexulose-6-phosphate synthase (EC:4.1.2.43, 5.3.1.27) of *Methanosaeta thermophila* that is present in the pentose phosphate pathway and methane metabolisms; or the 3-isopropylmalate dehydratase (EC:4.2.1.33, 4.2.1.35) of *Methanohalobium evestigatum* that is present in valine, leucine and isoleucine biosynthesis and C5-branched dibasic acid metabolisms. In addition, promiscuous enzymes could act in the same metabolic pathway, but catalyzing different steps, such as the phosphomethylpyrimidine kinase (EC:2.7.1.49, 2.7.4.7, 2.5.1.3) of *Thermococcus sibiricus*, that is involved in three different metabolic steps in thiamine metabolism; the ornithine acetyltransferase (EC:2.3.1.35, 2.3.1.1) of *Methanococcoides burtonii* that catalyzes two different activities in the cyclic version of arginine biosynthesis; and the polyprenyl synthetase (EC:2.5.1.1, 2.5.1.10, 2.5.1.29) of *Pyrobaculum aerophilum* that catalyzes three different steps in terpenoid backbone biosynthesis.

### 3.4. The Distribution of Promiscuous Enzymes between Functional Metabolic Categories is Variable

To determine the distribution of the promiscuous enzymes in the metabolism of archaeal genomes, each enzyme was mapped into the metabolic pathways defined by the KEGG database. At first, all enzymes (promiscuous and specialist) were associated with their metabolic pathways, and clustered into functional categories. Based on this data, the Carbohydrate Metabolism, Energy Metabolism and Metabolism of Cofactors and Vitamins were the three greater enrichment categories with promiscuous enzymes (Figure 3), while the Amino Acid Metabolism, Genetic Information Processing and Nucleotide Metabolism were the greater enrichment categories with specialist enzymes (Figure 3).

These findings suggest that the metabolic functions associated with the conservation and distribution of genetic information have evolved to recruit specialist enzymes to constrain the biochemical reactions towards the use of one substrate or one biochemical transformation, ensuring fidelity of information when being transmitted to the offspring, and forming part of the core of metabolic reactions shared by all cellular organisms [25]. In contrast, the metabolism of Carbohydrates has evolved for recruitment of enzymes able to use a great variety of substrates or performs more than one biochemical reaction allowing them to explore and use different carbon sources, such as the inositol monophosphatase from *Archaeoglobus fulgidus* and *Methanococcus jannaschii* which exhibit Inositol monophosphatase (IMPase), fructose-1,6-bisphosphatase and Nicotinamide adenine dinucleotide phosphate (NADP(H)) phosphatase activities [26,27], the phosphoglucose isomerase from *Pyrobaculum aerophilum* which catalyzes the isomerization of both glucose-6-phosphate and mannose-6-phosphate to fructose-6-phosphate [28], or the glucose dehydrogenase from *Sulfolobus solfataricus* with a broad range of sugar substrates, including d-glucose, d-galactose, d-xylose, and l-arabinose [29]. Regarding the latter, the promiscuity for the metabolism of glucose and galactose enables *S*. *solfataricus* to grow on both carbon sources and is a remarkable example of the unusual features and versatility of central carbohydrate metabolism in hyperthermophilic archaea [30].

In order to perform an analysis of the phylogenetic distribution of the archaeal promiscuous enzymes in the metabolism, we analyzed the enzymatic repertoire of Crenarchaeota and Euryarchaeota phyla. From this analysis we found that Carbohydrate and Energy Metabolisms are enriched in Crenarchaeota (Figure 4); where organisms of the order Sulfolobales, Desulforococcales, Thermoproteales and Acidilobales, mainly described as thermophilic, hyperthermophilic, and acidophilic are included. These organisms have different metabolic modes, ranging from chemoorganotrophs to chemolithotrophs [31], reinforcing the notion that the enriched metabolism found and previously described, correlates with the energy obtaining in Crenarchaeota, such as *S. solfataricus*, which grows aerobically and chemoorganotrophically obtained energy from organic sources such as sugars [32]. In counterpart, in Euryarchaeota, the metabolic categories related to Carbohydrates and Energy metabolisms are less enriched with promiscuous enzymes. Organisms included in this phylum, can only use a few simple substrates, most of them compounds such as H_2_/CO_2_, methanoate, methanol or methylamine, although any particular species is only capable of using two or three of them [33,34]. Or, they can use light as a source of energy as the organisms belonging to the class Haloarchaea, which have a pigment called bacteriorhodopsin that reacts with light producing a gradient of protons along the membrane that allows the synthesis of ATP [35]. Therefore, Euryarchaeota exhibit a lower metabolic plasticity in the use of substrates to obtain energy, probably due to a smaller enrichment of promiscuous enzymes in the metabolic categories of Carbohydrates and Energy. Finally, it seems that in the Euryarchaeota organisms, the metabolic processes associated with genetic information and the processes of protein synthesis have been favored to have a greater number of promiscuous enzymes (Figure 4).

### 3.5. Promiscuous Enzymes Exhibit a High Enrichment of Substrate Promiscuity rather than Catalytic Promiscuity

In recent years it has been seen that promiscuous enzymes are capable of carrying out more than one reaction or recognizing more than one substrate. In order to ascertain the extent of the types of promiscuity, both substrate and catalysis, promiscuous enzymes were classified based on their first and last digits of their E.C. numbers. Catalysis of the same reaction over a wide range of substrates is called substrate promiscuity, while when an enzyme catalyzes a different reaction for which it has been specialized with a different transition state is termed catalytic promiscuity [7,36]. In operational terms, enzymes exhibiting catalytic promiscuity are those that differ in the first digit from their E.C. numbers; while enzymes exhibiting substrate promiscuity are those that differ in the last digit from their associated E.C. numbers. The average number of enzymes with substrate promiscuity identified in Archaea is 29.7, while the average number of enzymes with catalytic promiscuity is 11.5, presenting a statistically significant difference (Wilcoxon test, *p*-value = 2.2 × 10^−16^). To carry out the comparison of promiscuity classes in the enzymatic repertoire of archaeal genomes, the ratio between enzymes with catalytic or substrate promiscuity among the total promiscuous enzymes present in each genome was calculated. From this normalization, an enrichment of enzymes with substrate promiscuity in comparison to enzymes with catalytic promiscuity was observed, with an average value of 70% and 29%, respectively, i.e., 2.5 times more of substrate promiscuity than catalytic promiscuity (Wilcoxon test, *p*-value = 2.2 × 10^−16^). In summary, almost three quarters of the promiscuous enzymes exhibit substrate promiscuity, while almost one-third show catalytic promiscuity, suggesting that it is more difficult to evolve towards a new catalytic mechanism than a new substrate-binding site [22].

### 3.6. Substrate Promiscuity is Represented to a Greater Extent in the Functional Metabolic Categories

To determine the distribution of the type of promiscuity in the metabolism of archaeal genomes, each promiscuous enzyme was mapped into the metabolic pathways defined by KEGG database. At first, all promiscuous enzymes (substrate or catalytic promiscuity) were associated with their metabolic pathways, and then were clustered into functional metabolic categories. Based on this data, we found that enzymes with substrate promiscuity are present to a greater extent in all functional metabolic categories; in contrast, enzymes with catalytic promiscuity are less represented in the metabolic pathways (Figure 5). This finding suggests that in the evolution of the promiscuous enzymes it is easier to modify the region of proteins that bind and recognize the substrate than the catalytic region that performs the biochemical transformations, as previously suggested [22].

For instance, in the Carbohydrate Metabolism category, the metabolic plasticity that provides the promiscuity of the substrate enables the exploration of a wide variety of compounds that can be used to generate energy. It is interesting to note that Nucleotide metabolism has a minor variation of catalytic functions, i.e., the number of enzymes with catalytic promiscuity is smaller, as was previously observed [22], in which was found a greater conservation of chemistry of the specialist enzymes involved in the nucleotide metabolism.

In this regard, to determine the distribution of the type of promiscuity in the metabolic categories in relation to phylogenetic taxonomy, all organisms were grouped into the main phyla Euryarchaeota and Crenarchaeota, and the average fraction of promiscuous enzymes was calculated. From this data, the content of enzymes with substrate promiscuity is not different in Euryarchaeota and Crenarchaeota, which have average values of 70% and 69%, respectively (Wilcoxon test, *p*-value = 0.39). Similarly, the fraction of enzymes with catalytic promiscuity in Euryarchaeota and Crenarchaeota is not different, with average values 28.38% and 28.57%, respectively (Wilcoxon test, *p*-value = 0.84). When the types of promiscuity are mapped on their respective metabolic pathways, and grouped into their functional metabolic categories, it was observed that the enzymes with substrate promiscuity belonging to Crenarchaeota are better represented in the Carbohydrate and Energy Metabolism categories (Table 2). In contrast, enzymes with catalytic promiscuity belonging to the Euryarchaeota are better represented in the Metabolism of Cofactors and Vitamins and Genetic Information Processing categories (Table 2). In summary, these findings suggest that although promiscuous enzymes are widely distributed in Euryarchaeota and Crenarchaeota phyla, there is a slight preference to some metabolic pathways.

Finally, an exhaustive comparison on their promiscuous enzymes between all archaeal organisms (see Appendix A), evidenced the existence of diverse metabolic pathways where these proteins are overrepresented and associated with archaeal lineages; such as methane, pentose phosphate, C5-branched dibasic acid, valine/leucine/isoleucine biosynthesis, porphyrin and chloropyll metabolisms associated with Euryarchaeota. In contrast, oxidative phosphorylation and nitrogen metabolisms are enriched with promiscuous enzymes in Crenarchaeota. In addition, diverse metabolic pathways enriched with promiscuous enzymes are not associated with any specific phylum, such as synthesis and degradation of ketone bodies, lysine degradation, fatty acid, benzoate degradation, and tryptophan among others, identified in genomes of Euryarchaeota *(Haloarcula marismortui*, *Haloferax volcanii*, *Natrialba magadii*, *Haloterrigena turkmenica*), and Crenarchaeota (*Sulfolobus)*. Therefore, the presence of promiscuous enzymes in Archaea correlates more preferentially with the metabolic pathways than life-styles, such as it was observed with *Sulfolobus*, where sulfur metabolism is almost absent of promiscuous enzymes, suggesting the prevalence of specific enzymes.

## 4. Conclusions

Through the analysis of 89 archaeal genomes, we gained insights into how many promiscuous enzymes there are, what the existing diversity of structural domains is, and how promiscuous enzymes are distributed in metabolic pathways of Archaea organisms. Although our criterion for the identification of enzymes was stringent, because it considers the association of each protein sequence to structural (Superfamily) domain and the annotation of E.C. numbers, the resulting dataset analyzed allowed us to obtain general trends that reflect the metabolic context of archaeal organisms. In this regard, we found that in spite of having a smaller fraction of promiscuous enzymes in the archaeal genomes, its structural diversity is greater than that of the specialized enzymes, which may contribute to participate in more than one metabolic pathway, or in more than one metabolic step within the same pathway. Another interesting finding of our analysis is an enrichment of substrate promiscuity rather than catalytic promiscuity in archaeal promiscuous enzymes, which suggests that in the evolution of the promiscuous enzymes it is easier to modify the region of proteins that bind and recognize the substrate than the catalytic region that performs the biochemical transformations. We also found that exists a heterogeneous distribution of promiscuous enzymes in archaeal genomes, favoring the enrichment of promiscuous enzymes in metabolic pathways associated with the generation of energy in Crenarchaeota organism, whereas in Euryarchaeota organisms there is a greater enrichment of promiscuous enzymes in metabolic pathways associated with genetic information and protein synthesis. Existence of promiscuous enzymes in metabolic pathways may be an adaptation to survival in its extreme environment allowing it to efficiently scavenge for energy substrates [37]. Our observations can be considered valid for all archaeal species and their metabolic pathways, and the results reported here increase our understanding in the evolution of the metabolism of archaeal organisms and in particular the role of promiscuous enzymes.

## Figures and Tables

**Figure 1 life-07-00030-f001:**
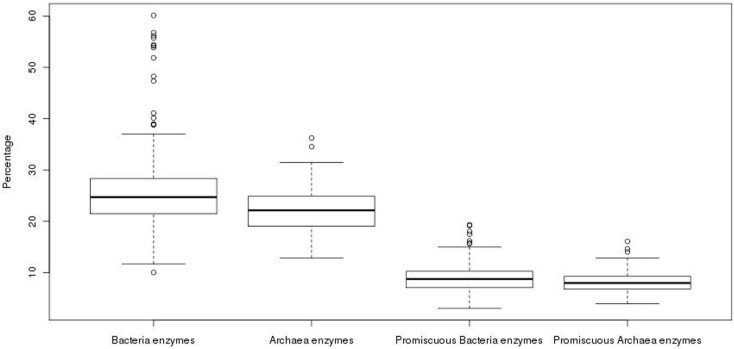
Proportion of enzymes in prokaryotic organisms. The proportion of total and promiscuous enzymes in bacterial and archaeal genomes is shown.

**Figure 2 life-07-00030-f002:**
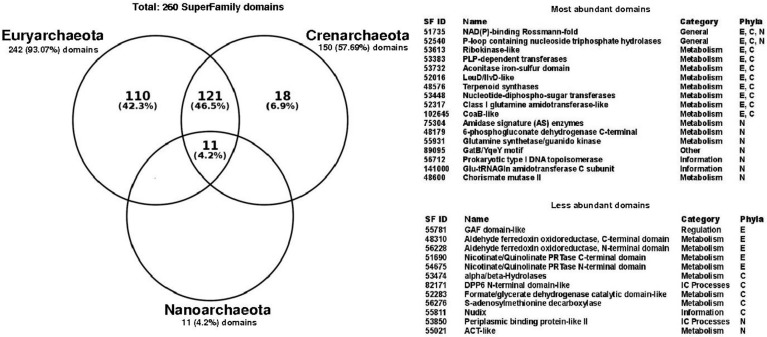
Structural domains associated with promiscuous enzymes in Archaea. Venn diagram shows the intersections of Superfamily domains corresponding to Euryarchaeota (242), Crenarchaeota (150) and Nanoarchaeota (11). Seventeen most abundant domains and twelve less abundant domains are showed by their Superfamily ID, name, functional category and phyla. E: Euryarchaeota; C: Crenarchaeota; N: Nanoarchaeota. Abbreviations used in this table: CoaB: 4′-Phosphopantothenoylcysteine synthetase; PRTase: Phosphoribosyl transferase; Nudix: Nudix stands for **Nu**cleoside **Di**phosphate linked to **X**; ACT-like: The ACT domain is named after three of the proteins that contain it **A**spartate kinase, **C**horismate mutase, and **T**yrA; NAD(P): Nicotinamide adenine dinucleotide phosphate; PLP: Pyridoxal phosphate; Glu-tRNAGln: Glutamyl-tRNA (Gln) amidotransferase; GAF domains-like: c**G**MP-specific phosphodiesterases, **a**denylyl cyclases and **F**hlA; DPP6: Dipeptidyl peptidase.

**Figure 3 life-07-00030-f003:**
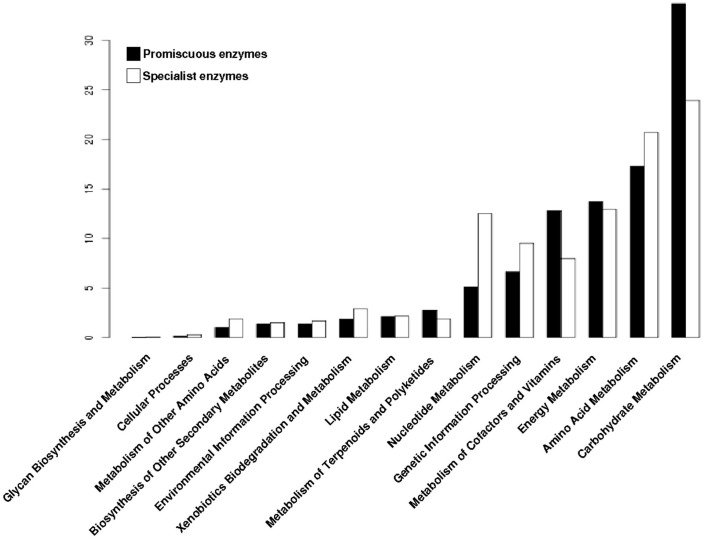
Distribution of promiscuous and specialist enzymes of archaeal genomes into metabolic categories of Kyoto Encyclopedia of Genes and Genomes (KEGG). The proportion of the two types of enzymes, promiscuous and specialist, into metabolic categories of KEGG.

**Figure 4 life-07-00030-f004:**
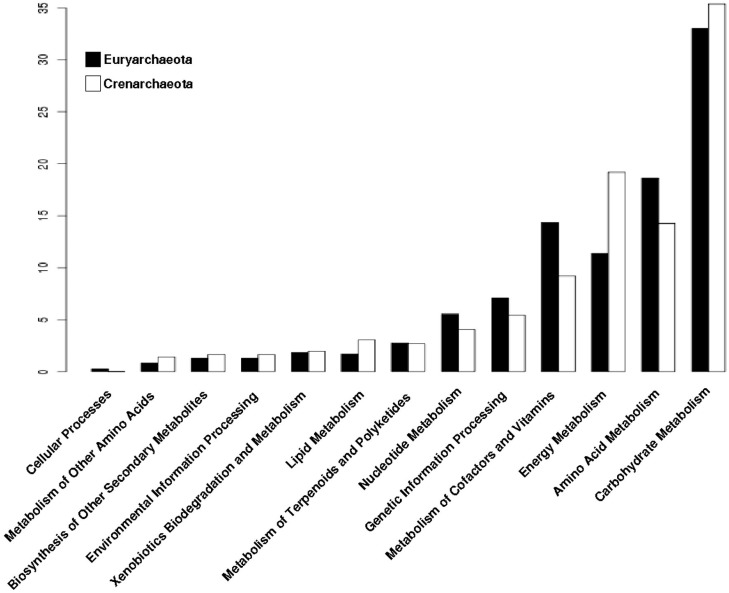
Distribution of promiscuous enzymes of archaeal genomes by phylum into metabolic categories of KEGG. Archaeal genomes were separated by phylum and their proportion of promiscuous enzymes is shown into metabolic categories of KEGG.

**Figure 5 life-07-00030-f005:**
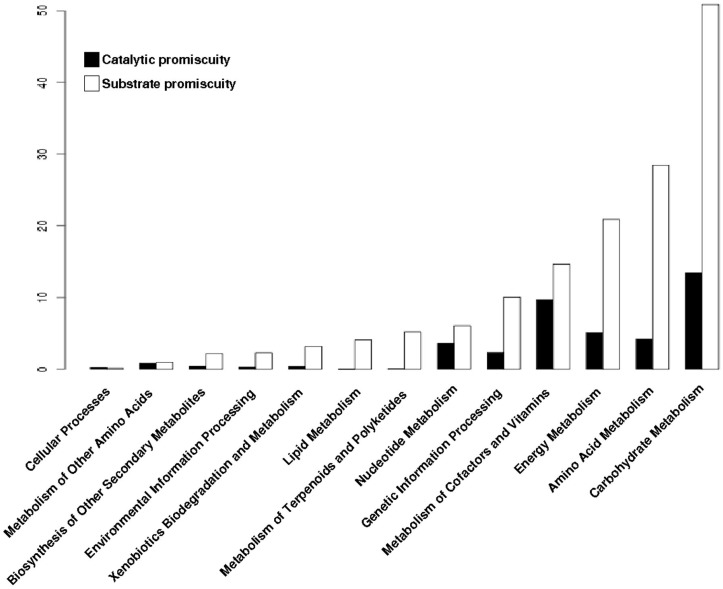
Distribution of enzymes by their type of promiscuity. Proportion of substrate or catalytic promiscuity of archaeal enzymes into metabolic categories of KEGG.

**Table 1 life-07-00030-t001:** Unique Superfamily structural domains found in archaeal promiscuous enzymes. In brackets is indicated the total of genomes where the domain was identified.

Superfamily	Supfam ID	Functional Annotation	Phylum
Sulfolobus fructose-1,6-bisphosphatase-like superfamily (66)	111249	Metabolism	Euryarchaeota Crenarchaeota
Glu-tRNAGln amidotransferase C subunit superfamily (56)	141000	Information Translation	Euryarchaeota Crenarchaeota Nanoarchaeota
Siroheme synthase middle domains-like superfamily (51)	75615	Metabolism	Euryarchaeota Crenarchaeota
Heme-dependent peroxidases superfamily (18)	48113	Metabolism Redox	Euryarchaeota
Nitrous oxide reductase, *N*-terminal domain superfamily (4)	50974	Metabolism Redox	Euryarchaeota Crenarchaeota
Oxidoreductase molybdopterin-binding domain superfamily (3)	56524	Metabolism E- transfer	Euryarchaeota
Tropomyosin superfamily (2)	57997	Intra-Cellular processes Cell motility	Euryarchaeota
DNA breaking-rejoining enzymes superfamily (2)	56349	Information DNA replication/repair	Euryarchaeota Crenarchaeota
*N*-terminal domain of MutM-like DNA repair proteins superfamily (2)	81624	Information DNA replication/repair	Euryarchaeota
(Phosphotyrosine protein) phosphatases II superfamily (2)	52799	Regulation Kinases/phosphatases	Crenarchaeota
6-hydroxymethyl-7,8-dihydropterin pyrophosphokinase, HPPK (7,8-dihydro-6-hydroxymethylpterin-pyrophosphokinase) superfamily (1)	55083	Metabolism Coenzyme metabolism and transport	Crenarchaeota
Eukaryotic DNA topoisomerase I, *N*-terminal DNA-binding fragment superfamily (1)	56741	Information DNA replication/repair	Crenarchaeota

**Table 2 life-07-00030-t002:** Distribution of promiscuous enzymes into metabolic categories of KEGG.

Metabolic Categories (KEGG)	Substrate Promiscuity (%)		Catalytic Promiscuity (%)	
	Euryarchaeota	Crenarchaeota	Euryarchaeota	Crenarchaeota
Carbohydrate metabolism	33.21	36.12	32.26	34.79
Amino acid metabolism	21.06	14.69	10.36	9.94
Energy metabolism	11.87	18.81	10.04	19.59
Metabolism of cofactors and vitamins	10.94	7.34	25.96	17.83
Genetic information processing	7.04	5.9	6.94	2.63
Nucleotide metabolism	4.62	2.88	8.86	9.35
Metabolism of terpenoids and polyketides	3.53	3.36	0.21	0
Lipid metabolism	2.23	3.84	0	0
Xenobiotics biodegradation and metabolism	1.98	2.47	1.38	0
Environmental information processing	1.33	2.06	0.96	0
Biosynthesis of other secondary metabolites	1.61	1.23	0.21	3.5
Metabolism of other amino acids	0.4	1.23	2.02	2.33
Cellular Processes	0.12	0	0.74	0

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
