# Peer review of "Tracing the Repertoire of Promiscuous Enzymes along the Metabolic Pathways in Archaeal Organisms"

_life, 2017, doi:10.3390/life7030030_

Round 1

Reviewer 1 Report

Here are my comments to Martinez-Nunez et al. This study is an attempt to characterize the the occurrence of promiscuous enzymes in Archea in comparison to Bacteria, and then addresses the differences between substrate and catalytic promiscuity within Archea. The paper reads nicely, is important, and addresses an understudied problem. In particular I like that the authors follow a very systematic approach,  so even though the study is only descriptive, one may learn a lot. At the same time however, the paper has major flaws in the sense that it does not correct, nor define, nor even attempts to alleviate or at least name the testing biases that exist in the datasets used to derive their conclusions. I give here some examples, but the scheme is throughput the paper

Result that genomic enzyme content is 1.5 times lower in Archea. Less protein work is done on Archaea as on bacteria, and  hence a homology search use to annotate the genomes will overrate the bacteria over archean enzyme content perhaps. Is there an algorithm available that can detect enzymes in an unbiased fashion; maybe  de novo gene prediction ? Otherwise the Authors would need to re-write this,,,, i,e, 1.5 times less known enzymes. I have one idea how the authors could test for this. Take instead of all, only those Archean and Bacterial proteins for which at least, let's say, five papers have been published. Is there still a difference? Then extrapolate for total genome on this basis. I would be happy with other approaches, but I think the authors need to address such bias.

If it holds true however… the question is why? Habitat perhaps? If this is true, thermophilic Archea might perhaos differ in enzyme content from mesophilic Archea, as more non-enzymatic reactions could replace enzymatic one.

For the same reason, I’m not convinced that the borderline significant, and numerically very small difference between the number of promiscuous enzymes between Archaea and Bacteria is any real. Again, there might be a testing bias as bacterial enzymes have been more studied. Also this could be ruled out by a well-studied subset. One can check how many publications are about each EC class, i.e. in the Brenda database. If the authors instead of all enzymes, include only the top studied enzymes, the difference should get larger if real, and much smaller if caused by a testing bias.

Same argument applies to the analysis of substrate versus catalytic promiscuity. Here I tend to agree with the authors with the conclusions. But also here a ‘well-studied subset’ analysis would help a lot to create confidence in the result.

I suggest it is imperative to a) try to address these biases and b) to discuss them broadly.

Finally I'm surprised by the, in difference to my expectations, very low overlap between protein superfamily contents between the Archean lineages. I have not repeated the analysis and I'm not claiming its not correct, but I would recommend the authors to have a self-critical double-check on this. Any cut-off problems that might explain the results perhaps?

Author Response

Here are my comments to Martinez-Nunez et al. This study is an attempt to characterize the occurrence of promiscuous enzymes in Archea in comparison to Bacteria, and then addresses the differences between substrate and catalytic promiscuity within Archea. The paper reads nicely, is important, and addresses an understudied problem. In particular I like that the authors follow a very systematic approach, so even though the study is only descriptive, one may learn a lot. At the same time however, the paper has major flaws in the sense that it does not correct, nor define, nor even attempts to alleviate or at least name the testing biases that exist in the datasets used to derive their conclusions. I give here some examples, but the scheme is throughput the paper.

Result that genomic enzyme content is 1.5 times lower in Archea. Less protein work is done on Archaea as on bacteria, and hence a homology search use to annotate the genomes will overrate the bacteria over archean enzyme content perhaps. Is there an algorithm available that can detect enzymes in an unbiased fashion; maybe de novo gene prediction ? Otherwise the Authors would need to re-write this,,,, i,e, 1.5 times less known enzymes. I have one idea how the authors could test for this. Take instead of all, only those Archean and Bacterial proteins for which at least, let's say, five papers have been published. Is there still a difference? Then extrapolate for total genome on this basis. I would be happy with other approaches, but I think the authors need to address such bias.

If it holds true however… the question is why? Habitat perhaps? If this is true, thermophilic Archea might perhaos differ in enzyme content from mesophilic Archea, as more non-enzymatic reactions could replace enzymatic one.

RESPONSE:  We understand that there are diverse highly annotated genomes (experimental evidences or global expression data) in contrast with genomes where annotations have been derived from sequence comparisons. In this regard, it is difficult to determine the quality of annotations for all the bacterial and archaeal genomes, representing an open problem in genomics (Schnoes et al., 2009). 

The annotation of bacterial and archaeal genomes is a process that is not only performed using a sequence-level homology search criterion. The annotated proteomes were taken from the NCBI database, whose annotation processes of both Bacteria and Archaea “is a multi-level process that includes prediction of protein-coding genes, as well as other functional genome units such as structural RNAs, tRNAs, small RNAs, pseudogenes, control regions, direct and inverted repeats, insertion sequences, transposons and other mobile elements. ... prokaryotic genome annotation pipeline ... combines ab initio gene prediction algorithms with homology based methods” (Tatusova et al. 2013). These criteria different than sequence homology seek to diminish possible bias to underestimate or overrated gene annotation in prokaryotic genomes. 

With regard to the annotation of sequences as enzymes, we have used the KEGG database, which takes as a starting point the annotations made by GenBank. Currently the KEGG database in addition to using annotations made by GenBank, uses functional orthologs to define enzymes, named as KEGG Orthology (KO). In the KO database, each KO entry is associated with references and sequence data of experimentally characterized proteins, so that the validity of sequence similarity-based extension can be examined for each ortholog group (Kanehisa, M. 2017). Molecular-level functions are stored in the KO database and associated with ortholog groups in order to enable extension of experimental evidence in a specific organism to other organisms. In this way, the identification of enzymes by KEGG in a genome takes into account not only the homology of the sequence, but also goes through a process of validation or healing of the annotations. 

In this regard, we have modified the paragraph to clarify the possible bias in the identification of the enzymes in the analyzed genomes, so that now it has been as follows:

In order to compare the total repertoire of known enzymatic proteins in the genomes of the Archaea, the content of these proteins was exhaustively evaluated … the average content of known enzymes in archaeal genomes is 1.5 times lower than the observed in bacterial genomes … When these values were normalized to obtain the percentage of known enzymes with respect to all the genes of each genome … with a higher proportion of known enzymes in Bacteria with around one-quarter of their genes encode for enzymes than in Archaea, the latter with one-fifth of the genes encode for enzymes … To evaluate whether there is an increase in the number of promiscuous enzymes in the repertoire of known total enzymes previously described”. 

As the reviewer comments, the amount of information associated with proteins in genomes is not the same at all. The largest amount of data available is concentrated in a few model organisms such as Escherichia coli or Bacillus subtilis.  In this regard, we made the exercise proposed by the reviewer with the data from RegulonDB database (Gama-Castro et al. 2016) of taking only those proteins that had 5 papers published in the model organism E. coli. The number of enzymatic proteins that meet this criterion is 124 out of a total of 4679 genes present in this organism. This represents a reduced number of enzymes, which would be much smaller for those genomes that are not model organisms. So it would be very difficult to recover an appropriate number of enzymes to be able to analyze them in those organisms for which there is little or no available information, and the differences that will be observed will be due to a bias in the available information of each organism.

For the same reason, I’m not convinced that the borderline significant, and numerically very small difference between the number of promiscuous enzymes between Archaea and Bacteria is any real. Again, there might be a testing bias as bacterial enzymes have been more studied. Also this could be ruled out by a well-studied subset. One can check how many publications are about each EC class, i.e. in the Brenda database. If the authors instead of all enzymes, include only the top studied enzymes, the difference should get larger if real, and much smaller if caused by a testing bias.

Same argument applies to the analysis of substrate versus catalytic promiscuity. Here I tend to agree with the authors with the conclusions. But also here a ‘well-studied subset’ analysis would help a lot to create confidence in the result.

I suggest it is imperative to a) try to address these biases and b) to discuss them broadly.

RESPONSE: Already in 2010 Carbonell and Faulon have developed methods to identify promiscuous enzymes using a graph-based representation known as molecular signature, and had an accuracy of 85%, which is good. The promiscuous enzyme sequences identified in our study were evaluated using the Promis Server (Carbonell P., Faulon JL. 2010), as described in the Methods section of our manuscript. Therefore, we compared our data set (400 from out 51572 promiscuous sequences), and the manually cured dataset kindly provided by Carbonell and Faulon (400 from out 19411 promiscuous enzymes), with the Promis Server. In addition, a subset of 400 out 48846 non-promiscuous enzymes (negative control), also provided by Carbonell and Faulon, were also evaluated with the Promis server. From this analysis we found an accuracy of 63% to our dataset, against 69% to the Faulon’s dataset, suggesting that our definition of promiscuous enzymes is in general enough informative, and all results are significant. This information is included in the manuscript.

Finally I'm surprised by the, in difference to my expectations, very low overlap between protein superfamily contents between the Archean lineages. I have not repeated the analysis and I'm not claiming its not correct, but I would recommend the authors to have a self-critical double-check on this. Any cut-off problems that might explain the results perhaps?

RESPONSE: The low overlapping between domain superfamilies is because only the unique domains present in promiscuous enzymes were counted, and not all domains of all proteins of the genomes. If we counted the intersection between the domains of all the proteins of each phyla, the number would be greater. Speaking only of the unique structural domains associated with promiscuous enzymes, almost half of them (46.5%) are shared between organisms belonging to Euryarchaeota and  Crenarchaeota lineages, which is not a low number. The cut that was put was to account only to the unique domains present in promiscuous enzymes, and not to the one of all the enzymes or proteins of the genomes.

Schnoes, A.M., Brown, S.D., Dodevski, I. and Babbitt, P. C. (2009). Annotation error in public databases: misannotation of molecular function in enzyme superfamilies. PLoS Comput Biol. 5(12): e1000605.

Tatusova T, DiCuccio M, Badretdin A, et al. Prokaryotic Genome Annotation Pipeline. 2013 Dec 10. In: The NCBI Handbook. 2nd edition. Bethesda (MD): National Center for Biotechnology Information (US); 2013-. Available from: https://www.ncbi.nlm.nih.gov/books/NBK174280/.

Kanehisa, M. 2017. Enzyme Annotation and Metabolic Reconstruction Using KEGG. Methods mol Biol. 1611:135-145.

Gama-Castro et al. 2016. RegulonDB version 9.0: high-level integration of gene regulation, coexpression, motif clustering and beyond. Nucleic Acids Res. 44(D1):D133-43.

Carbonell P., Faulon JL. 2010. Molecular signatures-based prediction of enzyme promiscuity. 26(16):2012-9.

Reviewer 2 Report

This is an interesting in silico analysis of archaeal and bacterial genome sequences aimed at identifying promiscuous enzymes in a global genomic context. The most interesting findings are the differences in abundances of promiscuous and specialist enzymes in relation to a number of functional and phylogenetic categories within the Archaea. Why the authors chose to focus solely on Archaea is not well rationalized, as it would be equally intriguing to see if the distribution of promiscuous versus specialist enzymes by category and phylotype is also different (and metabolically tractable) across the Bacteria.

The authors’ unrationalized focus on the Archaea is particularly striking in Fig. 1 where, even though there is a statistical difference in the proportion of promiscuous enzymes between the two domains, the actual differences are very, very minute. This would argue that the distribution of promiscuous versus specialist enzymes is not particularly robust for differentiating genomes of the Archaea from the Bacteria, but rather that there is an interesting evolution towards promiscuous enzymes within certain functional categories of protein-coding sequences or even among specific phylotypes. The manuscript would be more compelling if both domains were examined with the same criteria to show whether there really is a domain difference (a good hypothesis to test!). It might even be worthwhile to not include the analysis of bacterial genomes at all and instead publish a second study asking the same questions within the Bacteria and then comparing that data set to the Archaea-only data set. There are several conclusions based on broad assumptions about Archaea versus Bacteria that are not particularly robust (i.e. Archaea are more inclined towards extreme environments). There are plenty of Bacteria in many extreme environments and plenty of Archaea in mild environments. Analysis of the Bacteria might show very similar distributions of promiscuous enzymes based on very similar physiological criteria. However, this hypothesis was not tested in the present work and likely should be to substantiate any claims regarding evolution of pathways within a single domain but not in the other.

1)   There are a number of acronyms without definition when describing the functional domain categories. Consider adding an appendix of abbreviations.

2)   What is meant by “betweenness” of metabolic networks? Is this a bonafide scientific term? Perhaps there can be a small definition added with first usage. Presumably, it’s an indication of enzymes with the same E.C. number occupying multiple pathways, but that’s a guess without reading the reference.

3)   Line 298-300 – isn’t this explanation fairly obvious based on principles of enzyme structure and kinetics? It seems like a stronger conclusion can be made – that as shown throughout biochemistry, substrate promiscuity is far more prevalent than catalytic promiscuity due to kinetic constraints of enzyme structures.

4)   Data in Figures 6 & 7 would be better represented in a table to more easily compare distributions of enzymes across the KEGG categories and phylotypes.

5)   There could be a better analysis between the Crenarchaeota and Euryarchaota, particularly when it comes to differences in cofactors. Presumably, the genomes of the Euryarchaeaota contain a number of methanogens that require many cofactors. This could easily contribute to a data skew across categories. Greater in-depth comparisons across archaeal phylotypes can be made if the authors decide to omit the comparison to bacterial genomes in this study.

6)   Line 343 needs to mention the comparison to the bacterial genomes if both data sets are to be included in this single study.

7)   There are several sentences that should be edited for proper English grammar and syntax (e.g. line 332- “founds” should be findings)

Author Response

This is an interesting in silico analysis of archaeal and bacterial genome sequences aimed at identifying promiscuous enzymes in a global genomic context. The most interesting findings are the differences in abundances of promiscuous and specialist enzymes in relation to a number of functional and phylogenetic categories within the Archaea. Why the authors chose to focus solely on Archaea is not well rationalized, as it would be equally intriguing to see if the distribution of promiscuous versus specialist enzymes by category and phylotype is also different (and metabolically tractable) across the Bacteria.

The authors’ unrationalized focus on the Archaea is particularly striking in Fig. 1 where, even though there is a statistical difference in the proportion of promiscuous enzymes between the two domains, the actual differences are very, very minute. This would argue that the distribution of promiscuous versus specialist enzymes is not particularly robust for differentiating genomes of the Archaea from the Bacteria, but rather that there is an interesting evolution towards promiscuous enzymes within certain functional categories of protein-coding sequences or even among specific phylotypes. The manuscript would be more compelling if both domains were examined with the same criteria to show whether there really is a domain difference (a good hypothesis to test!). It might even be worthwhile to not include the analysis of bacterial genomes at all and instead publish a second study asking the same questions within the Bacteria and then comparing that data set to the Archaea-only data set. There are several conclusions based on broad assumptions about Archaea versus Bacteria that are not particularly robust (i.e. Archaea are more inclined towards extreme environments). There are plenty of Bacteria in many extreme environments and plenty of Archaea in mild environments. Analysis of the Bacteria might show very similar distributions of promiscuous enzymes based on very similar physiological criteria. However, this hypothesis was not tested in the present work and likely should be to substantiate any claims regarding evolution of pathways within a single domain but not in the other.

RESPONSE: The reason why in the present study the content of promiscuous enzymes in the genomes of Archaea is treated almost exclusively, is because the work is directed to a special number on extremophile organisms, and a large part of Archaea organisms and their metabolic processes have such a characteristic, such as metanogens, chemoorganotrophs, chemolithotrophs. The aim of this study was not differentiate between Bacteria and Archaea organism using promiscuous enzymes, but to identify if in certain metabolic pathways or lineages there was a greater enrichment of promiscuous enzymes.

We appreciate the comments on the importance of including bacterial genomes in this analysis, but we have considered that such analysis should be done in a second study, as suggested by the reviewer.

1) There are a number of acronyms without definition when describing the functional domain categories. Consider adding an appendix of abbreviations.

RESPONSE: We have appropriately defined all acronyms described in the manuscript.

2) What is meant by “betweenness” of metabolic networks? Is this a bonafide scientific term? Perhaps there can be a small definition added with first usage. Presumably, it’s an indication of enzymes with the same E.C. number occupying multiple pathways, but that’s a guess without reading the reference.

RESPONSE: In effect, the “betweenness centrality” is a measure of how much of the metabolite traffic in the cell goes through a given pathway (Mazurie et al., 2010); however we have preferred exclude this definition from the manuscript to avoid confusion to the readers. Instead, we  provided examples of promiscuous enzymes acting on different metabolic pathways, or on the same metabolic pathway but catalyzing different steps. We emphasize that the structural diversity identified in promiscuous enzymes allows these enzymes to participate in more than one metabolic pathway, or in more than one metabolic step within the same pathway. We have provided some examples of promiscuous enzymes acting on different metabolic pathways, or on the same metabolic pathway but catalyzing different steps.

Mazurie, A.; Bonchev, D.; Schwikowski, B.; Buck, G.A. Evolution of metabolic network organization. BMC Syst Biol 2010, 4, 59.

3) Line 298-300 – isn’t this explanation fairly obvious based on principles of enzyme structure and kinetics? It seems like a stronger conclusion can be made – that as shown throughout biochemistry, substrate promiscuity is far more prevalent than catalytic promiscuity due to kinetic constraints of enzyme structures.

RESPONSE: It was proposed that the entire central metabolic pathway in Sulfolobus solfataricus is promiscuous for the metabolism of glucose and galactose. This situation is in contrast with other microorganisms, where separate enzymes and pathways are present for the metabolism of the two sugars (Milburn et al., 2006). Therefore, the substrate promiscuity of glucose dehydrogenase cannot be completely obvious from the structure of the enzyme or the kinetics of the enzyme. The conclusion that substrate promiscuity is far more prevalent than catalytic promiscuity is described in the manuscript and is dealt with in greater detail in the section 3.5 “Promiscuous enzymes exhibit a high enrichment of substrate promiscuity rather than catalytic promiscuity”.

Milburn, C. C.; Lamble, H. J.; Theodossis, A.; Bull, S. D.; Hough, D. W.; Danson, M. J.; Taylor, G. L. (2006) The Structural Basis of Substrate Promiscuity in Glucose Dehydrogenase from the Hyperthermophilic Archaeon Sulfolobus solfataricus. J Biol Chem. 281(21): 14796-804.

4) Data in Figures 6 & 7 would be better represented in a table to more easily compare distributions of enzymes across the KEGG categories and phylotypes.

RESPONSE: We agree with the reviewer and we have performed the representation of the data of Figures 6 and 7 in a table, allowing an easier comparison of the distribution of the promiscuous enzymes across the KEGG categories and phylotypes. Therefore, figures 6 and 7 were replaced by Table II.

5) There could be a better analysis between the Crenarchaeota and Euryarchaota, particularly when it comes to differences in cofactors. Presumably, the genomes of the Euryarchaeaota contain a number of methanogens that require many cofactors. This could easily contribute to a data skew across categories. Greater in-depth comparisons across archaeal phylotypes can be made if the authors decide to omit the comparison to bacterial genomes in this study.

RESPONSE: We have added the follow paragraph describing the most prominent results associated to Crenarchaeota and Euryarchaota comparison.

Finally, an exhaustive comparison on their promiscuous enzymes between all archaeal organisms (see supplementary material Figure S1), evidenced the existence of diverse metabolic pathways where these proteins are overrepresented and associated to archaeal lineages; such as methane, pentose phosphate, C5-branched dibasic acid, valine/leucine/isoleucine biosynthesis, porphyrin and chloropyll metabolisms associated to Euryarchaeota. In contrast, oxidative phosphorylation and nitrogen metabolisms are enriched with promiscuous enzymes in Crenarchaeota. In addition, diverse metabolic pathways enriched with promiscuous enzymes are not associated to any specific phylum, such as synthesis and degradation of ketone bodies, lysine degradation, fatty acid, benzoate degradation, and tryptophan among others, identified in genomes of Euryarchaeota (Haloarcula marismortui, Haloferax volcanii, Natrialba magadii, Haloterrigena turkmenica), and Crenarchaeota (Sulfolobus). Therefore, the presence of promiscuous enzymes in Archaea correlates preferentially with the metabolic pathways than life-styles, such as it was observed with Sulfolobus, where sulfur metabolism is almost absent of promiscuous enzymes, suggesting the prevalence of specific enzymes.

6) Line 343 needs to mention the comparison to the bacterial genomes if both data sets are to be included in this single study.

RESPONSE: In this study only archaeal genomes are considered, leaving for a further study the comparison of the average of promiscuous enzymes, both substrate and catalysis, in bacterial genomes.

7) There are several sentences that should be edited for proper English grammar and syntax (e.g. line 332- “founds” should be findings)
RESPONSE: We have edited and corrected all these sentences in the manuscript.

Reviewer 3 Report

Please add the title "Percentage" to the vertical axis in Fig. 3-7.

Author Response

p { margin-bottom: 0.1in; direction: ltr; color: rgb(0, 0, 10); line-height: 120%; text-align: left; }p.western { font-family: "Liberation Serif",serif; font-size: 12pt; }p.cjk { font-family: "AR PL SungtiL GB"; font-size: 12pt; }p.ctl { font-family: "FreeSans"; font-size: 12pt; }

Please add the title "Percentage" to the vertical axis in Fig. 3-7.

RESPONSE: We have added figure legends as reviewer suggested. The Figures 6 and 7 were suppressed and replaced by a table as suggested by reviewer 2.

Round 2

Reviewer 1 Report

The authors discuss the existing (literature) biases that may influence the conclusions of their study much better, have however done relatively little to address them.

Author Response

Comments and Suggestions for Authors

The authors discuss the existing (literature) biases that may influence the conclusions of their study much better, have however done relatively little to address them.

RESPONSE: In order to exclude any bias associated to the enzymatic repertoire, we have analyzed and selected all enzymes from KEGG database with the same criteria, i.e. those enzymes with more than two different E. C. numbers and SuperFamily domain annotations. We consider this criterion to be strict and make all data robust to analyze and obtain conclusions. In this regard, we compared our data against the identification de novo proposed by Carbonell and Faulon (2010), finding an accuracy of 63% to our dataset, against 69% to the Faulon’s dataset, suggesting that our definition of promiscuous enzymes is in general enough informative, and all results are significant. This information is included in the manuscript.

We consider that the de novo identification of promiscous enzymes must be achieved in all the genomes, however this approach must be analyzed carefully, such as the Carbonell and Faulon (2010) approach that contains almost 70% of accuracy and 30% of probable mispredictions. Therefore, we consider that the results and conclusions described in this work are enough robust and must provide a framework to future discussion in the field.

In addition, we understand that genome annotation is an unsolved problem and it is beyond the scope of this analysis. In this regard, the use of international annotation standards developed by NCBI in collaboration with sequencing centers, archival databases, and researchers, will increase the quality of databases, enabling researchers to make accurate biological discoveries. Several pieces of evidences are combined to assign confidence levels to particular annotations or to predict functions, such as sequence similarity, phylogenomic or genomic context, comparative genomics, and in many cases a combination of all of the above. NCBI initiated the Reference Sequence database to create a curated non-redundant set of sequences. The Kyoto Encyclopedia of Genes and Genomes (KEGG) orthology groups (KO) uses NCBI Reference Sequences (Klimke et al., 2011), and sequence homology as gold reference to annotate protein sequences.

Klimke et al., 2011. Solving the Problem: Genome Annotation Standards before the Data Deluge. Stand Genomic Sci. 5(1):168-193.  PMCID: PMC3236044.

Reviewer 2 Report

The changes to the manuscript have improved the presentation substantially. In particular, it is much easier to compare the data in the tables than in the figures.

Author Response

The changes to the manuscript have improved the presentation substantially. In particular, it is much easier to compare the data in the tables than in the figures.

RESPONSE: Thank you for your comments, they were very helpful in improving our manuscript